# Bounded Variation Separates Weak and Strong Average Lipschitz

**DOI:** 10.3390/e27090974

**Published:** 2025-09-18

**Authors:** Ariel Elperin, Aryeh Kontorovich

**Affiliations:** Computer Science Department, Ben-Gurion University, Beer Sheva 84105, Israel; ariel.elperin@gmail.com

**Keywords:** Lipschitz, variation, smooth average

## Abstract

We closely examine a recently introduced notion of average smoothness. The latter defined a *weak* and *strong* average-Lipschitz seminorm for real-valued functions on general metric spaces. Specializing to the standard metric on the real line, we compare these notions to bounded variation (BV) and discover that the weak notion is strictly weaker than BV while the strong notion is strictly stronger. Along the way, we discover that the weak average smooth class is also considerably larger in a certain combinatorial sense, which is made precise by the fat-shattering dimension.

## 1. Introduction

A function f:[0,1]→R is *L*-Lipschitz if $\left|f\left(x\right)-f\left(x^{\prime }\right)\right|\leq L|x-x^{\prime }|$ for all $x,x^{\prime }\in \left[0,1\right]$, and ${∥f∥}_{\mathrm{Lip}}$ is the smallest *L* for which this holds. If *f* has an integrable derivative, its *variation* V(f) is given by $V\left(f\right)=\int_{0}^{1}\left|f^{\prime }\left(x\right)\right|dx$ (the more general definition is given in (Equation 4)). Since $\left|f^{\prime }\left(x\right)\right|\leq {∥f∥}_{\mathrm{Lip}}$, we have the obvious relation $V\left(f\right)\leq {∥f∥}_{\mathrm{Lip}}$. No reverse inequality is possible: since for monotone *f*, we have V(f)=|f(0)−f(1)| [1], a function whose value increases from 0 to ε with a sharp “jump” in the middle can have an arbitrarily large *L* and arbitrarily small *V*.

Motivated by questions in machine learning and statistics, Ashlagi et al. [2] introduced two notions of “average Lipschitz” in general metric probability spaces: a weak one and a strong one (follow-up works extended these results to average Hölder smoothness [3,4]). For the special case of the metric space Ω=[0,1] equipped with the standard metric $\rho \left(x,x^{\prime }\right)=\left|x-x^{\prime }\right|$ and the uniform distribution *U*, their definitions are as follows. Both notions rely on the *local slope* of f:[0,1]→R at a point *x*, which is defined (and denoted) as follows:(1)$$\begin{array}{ccc} \Lambda_{f}\left(x\right) & = & \underset{x^{\prime }\in \left[0,1\right]\setminus \left\{x\right\}}{\sup}\frac{|f\left(x\right)-f\left(x^{\prime }\right)|}{|x-x^{\prime }|},\;x\in \left[0,1\right]. \end{array}$$ The *strong* and *weak* average smoothness of *f* are defined, respectively, by$$\begin{array}{l} {∥f∥}_{S}=E\left[\Lambda_{f}\left(X\right)\right],\\ {∥f∥}_{W}=W\left[\Lambda_{f}\left(X\right)\right]=\underset{t>0}{\sup}tU\left(\left\{x\in \Omega :\Lambda_{f}\left(x\right)\geq t\right\}\right), \end{array}$$
where *X* is a random variable distributed according to *U* on [0,1], E is the usual expectation, and W is the *weak $L_{1}$ norm* of the random variable *Z*:$$W\left[Z\right]=\underset{t>0}{\sup}tP\left(|Z|\geq t\right).$$ Both ${∥\cdot ∥}_{S}$ and ${∥\cdot ∥}_{W}$ satisfy the homogeneity axiom of seminorms (meaning that $∥\alpha f∥=\left|\alpha \right|\cdot ∥f∥$), and ${∥\cdot ∥}_{S}$ additionally satisfies the triangle inequality and hence is a true seminorm. The *weak $L_{1}$* norm satisfies the weaker inequality W[X+Y]≤2(W[X]+W[Y]) [5], which ${∥\cdot ∥}_{W}$ also inherits.

We now recall the definition of the variation of f:[a,b]→R: (2)$$\begin{array}{c} V_{a}^{b}\left(f\right)=\underset{a=x_{0}<x_{1}<x_{2}<\ldots <x_{n}\leq b}{\sup}\sum_{i=1}^{n}\left|f\left(x_{i}\right)-f\left(x_{i-1}\right)\right| \end{array}$$ (when a=0 and b=1, we omit these), as well as the Lipschitz and bounded variation function classes: $$\begin{array}{l} \mathrm{Lip}=\left\{f:\left[0,1\right]\to R;\;{∥f∥}_{\mathrm{Lip}}<\infty \right\},\\ \mathrm{BV}=\left\{f:[0,1]\to R;\;V(f)<\infty \right\}. \end{array}$$ The discussion above implies the (well-known) strict containment(3)$$\begin{array}{c} \mathrm{Lip}⊊\mathrm{BV}. \end{array}$$ In addition, we define the strong and weak average smoothness classes$$\begin{array}{l} {\bar{\mathrm{Lip}}}^{s}=\left\{f:\left[0,1\right]\to {R;\;∥f∥}_{S}<\infty \right\},\\ {\bar{\mathrm{Lip}}}^{w}=\left\{f:\left[0,1\right]\to {R;\;∥f∥}_{W}<\infty \right\}. \end{array}$$ By Markov’s inequality and the fact that the expectation is bounded by the supremum, we have$${∥f∥}_{W}\leq {∥f∥}_{S}\leq \underset{x\in \Omega }{\sup}\Lambda_{f}\left(x\right)={∥f∥}_{\mathrm{Lip}}$$
whence (4)$$\begin{array}{c} \mathrm{Lip}\subseteq {\bar{\mathrm{Lip}}}^{s}\subseteq {\bar{\mathrm{Lip}}}^{w}; \end{array}$$
all of these containments were shown to be strict in [2]. The containments in (Equation 6) and (Equation 9) leave open the relation between BV and ${\bar{\mathrm{Lip}}}^{s},{\bar{\mathrm{Lip}}}^{w}$, which we resolve in this work:

**Theorem 1.** 
*${\bar{\mathrm{Lip}}}^{s}⊊\mathrm{BV}⊊{\bar{\mathrm{Lip}}}^{w}$.*


We also provide a quantitative, finitary relation between these clases:

**Theorem 2.** 
*For any f:[0,1]→R, we have $\frac{1}{2}{∥f∥}_{W}\leq V\left(f\right)\leq {∥f∥}_{S}.$*


Finally, we recall the definition of the *fat-shattering dimension*, which is a combinatorial complexity measure of function classes of central importance in statistics, empirical processes, and machine learning [6,7]. Let *F* be a collection of functions mapping [0,1] to R. For γ>0, a set $S=\left\{x_{1},\ldots ,x_{m}\right\}\subset \left[0,1\right]$ is said to be γ-shattered by *F* if(5)$$\begin{array}{c} \underset{r\in R^{m}}{\sup}\;\min_{y\in {\left\{-1,1\right\}}^{m}}\;\underset{f\in F}{\sup}\;\min_{i\in [m]}\;y_{i}\left(f\left(x_{i}\right)-r_{i}\right)\geq \gamma . \end{array}$$ The γ-fat-shattering dimension, denoted by ${\mathrm{fat}}_{\gamma }\left(F\right)$, is the size of the largest γ-shattered set (possibly *∞*). It is known [8] that for F={f:[0,1]→R∣V(f)≤L}, we have ${\mathrm{fat}}_{\gamma }\left(F\right)=1+\left\lfloor \frac{L}{2\gamma }\right\rfloor$. This same bound holds for F={f:[0,1]→R∣Lipf≤L}.

Although the strong smoothness class has the same combinatorial complexity as the BV and Lipschitz classes, for weak average smoothness, this quantity turns out to be considerably greater:

**Theorem 3.** *For L>0, let $F_{W}=\left\{f:\left[0,1\right]\to {R;∥f∥}_{W}\leq L\right\}$ and $F_{S}=\left\{f:\left[0,1\right]\to {R;∥f∥}_{S}\leq L\right\}$. Then,**1.* *${\mathrm{fat}}_{\gamma }\left(F_{W}\right)=\infty$ whenever $\gamma \leq \frac{L}{6}$;**2.* *${\mathrm{fat}}_{\gamma }\left(F_{S}\right)=1+\left\lfloor \frac{L}{2\gamma }\right\rfloor$ for γ>0.* Notation.

We write $\left[n\right]:=\left\{1,\ldots ,n\right\}$ and use m(·) to denote the Lebesgue measure (length) of sets in R.

## 2. Proofs

We begin with a variant of the standard covering lemma.

**Lemma 1.** 
*For any sequence $s_{1},\ldots ,s_{n}$ of closed segments in R, there is a subsequence indexed by I⊆[n] such that for all distinct i,j∈I we have $s_{i}\cap s_{j}=\emptyset$ and $\sum_{i\in I}m\left(s_{i}\right)\geq \frac{1}{2}m\left(\overset{n}{\underset{i=1}{⋃}}s_{i}\right)$.*


**Proof.** We proceed by induction on *n*. Let G=([n],E) denote the intersection graph of the $s_{i}$: the vertices correspond to the segments and (i,j)∈E if $s_{i}\cap s_{j}\neq \emptyset$.Suppose that *G* contains a cycle, and let $s_{1}=\left[a_{1},b_{1}\right],\ldots ,s_{k}=\left[a_{k},b_{k}\right]$ be the segments in the cycle sorted by their right endpoint. Since $s_{1}\cap s_{k}\neq \emptyset$, we have $a_{k}\leq b_{1}$. If $a_{k-1}\geq a_{1}$, then $s_{k-1}\subseteq s_{1}\cup s_{k}$. Otherwise, $a_{k-1}<a_{1}$ and $s_{1}\subseteq s_{k-1}$. Either way, we have found a segment that is completely covered by the other vertices of *G*. After removing it, we obtain I⊆[n] of size n−1 with $\underset{i\in I}{⋃}s_{i}=\overset{n}{\underset{i=1}{⋃}}s_{i}$, so applying the inductive hypothesis on the segments in *I* yields the desired result. If *G* does not contain a cycle, then G=A∪B is bipartite, where A,B⊆[n] are disjoint and nonempty. Clearly, $m\left(\overset{n}{\underset{i=1}{⋃}}s_{i}\right)\leq m\left(\underset{i\in A}{⋃}s_{i}\right)+m\left(\underset{i\in B}{⋃}s_{i}\right)$, and thus $\max\left(m\left(\underset{i\in A}{⋃}s_{i}\right),m\left(\underset{i\in B}{⋃}s_{i}\right)\right)\geq \frac{1}{2}m\left(\overset{n}{\underset{i=1}{⋃}}s_{i}\right)$, so taking either I=A or I=B (which is possible since the segments inside each part are disjoint) yields the desired result. □

Next, we reduce the proof of Theorem 2 to the case of right-continuous monotone functions.

**Lemma 2.** 
*If for every right-continuous monotone function f:[0,1]→R we have ${∥f∥}_{W}\leq 2V\left(f\right)$, then the bound holds for all f:[0,1]→R. Furthermore, both inequalities are tight.*


**Proof.** We begin by observing that we can restrict our attention to monotone functions, since $T_{f}\left(x\right)=V_{f}\left(\left[0,x\right]\right)$ is monotone and has the same variation as *f*, but $\Lambda_{T_{f}}\left(x\right)=\underset{x^{\prime }\neq x}{\sup}\frac{|T_{f}\left(x\right)-T_{f}\left(x^{\prime }\right)|}{|x-x^{\prime }|}\geq \underset{x^{\prime }\neq x}{\sup}\frac{|f\left(x\right)-f\left(x^{\prime }\right)|}{|x-x^{\prime }|}$, which means ${∥T_{f}∥}_{W}\geq {∥f∥}_{W}$.Thus, the inequality ${∥T_{f}∥}_{W}\leq 2V\left(T_{f}\right)$ immediately implies that ${∥f∥}_{W}\leq {∥T_{f}∥}_{W}\leq 2V\left(T_{f}\right)=2V\left(f\right)$. If *f* is monotone, it can only have jump discontinuities. Let I⊂[0,1] denote the set of right discontinuities of *f*. Note that since *f* is monotone, *I* is at most countable. Define the modified version of *f* to be(6)$$\begin{array}{c} \tilde{f}\left(x\right)=\left\{\begin{array}{cc} f(x) & x\notin I\\ \lim_{\varepsilon \to 0^{+}}f\left(x+\varepsilon \right) & x\in I. \end{array}\right. \end{array}$$ Note that $\tilde{f}$ is monotone and right-continuous. It is not hard to see that if 0∉I, then $V\left(\tilde{f}\right)=V\left(f\right)$ and $\Lambda_{\tilde{f}}\left(x\right)=\Lambda_{f}\left(x\right)$ for all x∉I, which implies that ${∥f∥}_{W}={∥\tilde{f}∥}_{W}$ and allows us to restrict our discussion to right-continuous functions. If 0∈I, then we can extend the domain of $\tilde{f}$ to [−ε,1] for all ε>0, where $\tilde{f}\left(x\right)=f\left(0\right)$ for all x<0. Denote the extended function by ${\tilde{f}}_{\varepsilon }$, then since ${∥f∥}_{W}=\lim_{\varepsilon \to 0}{∥{\tilde{f}}_{\varepsilon }∥}_{W}$ and $V\left({\tilde{f}}_{\varepsilon }\right)=V\left(f\right)$ for all ε>0, we can conclude that$${∥f∥}_{W}=\lim_{\varepsilon \to 0}{∥{\tilde{f}}_{\varepsilon }∥}_{W}\leq \lim_{\varepsilon \to 0}2V\left({\tilde{f}}_{\varepsilon }\right)=2V\left(f\right).$$□

### 2.1. Proof of Theorem 2

We first show that ${∥f∥}_{W}\leq 2V\left(f\right)$. We may assume without loss of generality that V(f)<∞. We will use the notation $V_{f}\left(\left[a,b\right]\right)$ for the variation of *f* when restricted to the segment [a,b]. Since *f* is of bounded variation, the function $T_{f}\left(x\right)=V_{f}\left(\left[0,x\right]\right)$ is well defined for x>0. By Lemma 2, we may assume without loss of generality that *f* is right-continuous. Thus, $T_{f}:\left[0,1\right]\to R$ is monotone and right-continuous and thus induces a Lebesgue–Stieltjes measure on [0,1], which we denote by $\mu_{f}$. We now define the *maximal* function $M_{f}:\left[0,1\right]\to R$ as follows:(7)$$\begin{array}{c} M_{f}\left(x\right)=\underset{r_{1},r_{2}>0}{\sup}\frac{\mu_{f}\left(\left[x-r_{1},x+r_{2}\right]\right)}{r_{1}+r_{2}}=\underset{r_{1},r_{2}>0}{\sup}\frac{V_{f}\left(\left[x-r_{1},x+r_{2}\right]\right)}{r_{1}+r_{2}}, \end{array}$$
where the segments [a,x],[x,b] are taken to be [0,x],[x,1], respectively, whenever a<0 or b>1. A standard argument shows that $M_{f}^{-1}\left((t,\infty )\right)$ is open, whence $M_{f}$ is measurable.

We now observe that $M_{f}\geq \Lambda_{f}$ everywhere in [0,1]. Indeed, if $x^{\prime }>x$, then $M_{f}\left(x\right)\geq \frac{V_{f}\left(\left[x-\varepsilon ,x^{\prime }\right]\right)}{\varepsilon +(x^{\prime }-x)}\geq \frac{|f\left(x^{\prime }\right)-f\left(x\right)|}{x^{\prime }-x+\varepsilon }$ holds for ε>0, and hence $M_{f}\left(x\right)\geq \underset{x^{\prime }>x}{\sup}\frac{|f\left(x^{\prime }\right)-f\left(x\right)|}{x^{\prime }-x}$. The case of $x^{\prime }<x$ is completely analogous, whence $M_{f}\left(x\right)\geq \underset{x^{\prime }\neq x}{\sup}\frac{|f\left(x\right)-f\left(x^{\prime }\right)|}{|x-x^{\prime }|}=\Lambda_{f}\left(x\right)$. For *X* uniformly distributed over [0,1], we have $P\left(\Lambda_{f}\left(X\right)\geq t\right)\leq P\left(M_{f}\left(X\right)\geq t\right)$ and showing ${∥f∥}_{W}\leq 2V\left(f\right)$ reduces to bounding the latter probability by 2V(f)/t.

We now closely follow the proof of Theorem 7.4 in [9] and bound $m\left(M_{f}^{-1}\left((t,\infty )\right)\right)$ by bounding m(K) for arbitrary compact $K\subseteq M_{f}^{-1}\left((t,\infty )\right)$. For $x\in K\subseteq M_{f}^{-1}\left((t,\infty )\right)$, denote by $r_{1}\left(x\right),r_{2}\left(x\right)$ some lengths such that $\frac{\mu_{f}\left([x-r_{1}\left(x\right),x+r_{2}\left(x\right)]\right)}{r_{1}\left(x\right)+r_{2}\left(x\right)}\geq t$. Denote by $S_{x}$ the open interval $\left(x-r_{1}\left(x\right),x+r_{2}\left(x\right)\right)$. Then, clearly, $K\subseteq \underset{x\in K}{⋃}S_{x}$. Since *K* is compact, a finite cover by intervals $S_{x_{1}},\ldots ,S_{x_{n}}$ exists. By Lemma 1, there exists I⊆[n] such that for all distinct i,j∈I, we have $S_{x_{i}}\cap S_{x_{j}}=\emptyset$ and $\sum_{i\in I}m\left(S_{x_{i}}\right)\geq \frac{1}{2}m\left(\overset{n}{\underset{i=1}{⋃}}S_{x_{i}}\right)$. Finally, by the definition of the $S_{x}$’s, for each i∈[n], it holds that $m\left(S_{x_{i}}\right)\leq \frac{\mu_{f}\left(S_{x_{i}}\right)}{t}$. We can now write(8)$$\begin{array}{c} m\left(K\right)\leq m\left(\overset{n}{\underset{i=1}{⋃}}S_{x_{i}}\right)\leq 2\sum_{i\in I}m\left(S_{x_{i}}\right)\leq \frac{2}{t}\sum_{i\in I}\mu_{f}\left(S_{x_{i}}\right)\leq \frac{2}{t}\mu_{f}\left(\left[0,1\right]\right), \end{array}$$
where the last inequality holds since the intervals in *I* are disjoint. Since $\mu_{f}\left(\left[0,1\right]\right)=V\left(f\right)$, it immediately follows that ${∥f∥}_{W}\leq 2V\left(f\right)$.

It remains to show that $V\left(f\right)\leq {∥f∥}_{S}$. Let us denote by $P_{n}$ the partition $0\leq x_{1}<x_{2}<\ldots <x_{n}\leq 1$ of [0,1], and let $V\left(P_{n}\right)=\sum_{i=1}^{n-1}\left|f\left(x_{i+1}\right)-f\left(x_{i}\right)\right|$ denote the variation of *f* relative to $P_{n}$. It suffices to show that for any such partition $P_{n}$, we have ${∥f∥}_{S}\geq V\left(P_{n}\right)$. Now(9)$$\begin{array}{c} {∥f∥}_{S}=E\left[\Lambda_{f}\left(X\right)\right]\geq \sum_{i=1}^{n-1}\left|x_{i+1}-x_{i}\right|E\left[\Lambda_{f}\left(X\right)|X\in \left[x_{i},x_{i+1}\right]\right]. \end{array}$$

Note that for all $x\in [x_{i},x_{i+1}]$, we have(10)$$\begin{array}{c} \Lambda_{f}\left(x\right)\geq \max\left(\frac{|f\left(x\right)-f\left(x_{i}\right)|}{x-x_{i}},\frac{|f\left(x_{i+1}\right)-f\left(x\right)|}{x_{i+1}-x}\right)\geq \frac{|f\left(x_{i+1}\right)-f\left(x_{i}\right)|}{x_{i+1}-x_{i}}. \end{array}$$ Applying this to (Equation 15) yields$$E\left[\Lambda_{f}\right]\geq \sum_{i=1}^{n-1}\left|x_{i+1}-x_{i}\right|\frac{|f\left(x_{i+1}\right)-f\left(x_{i}\right)|}{x_{i+1}-x_{i}}=V\left(P_{n}\right).$$

Finally, the tightness of the first claimed inequality is witnessed by the step function f(x)=1[x>1/2] and of the second inequality by f(x)=x. □

### 2.2. Proof of Theorem 1

The claimed containments are immediate from Theorem 2; only the separations remain to be shown. The first of these is obvious: the step function has bounded variation but infinite strong average Lipschitz [2] (Appendix I). We proceed with the second separation:

**Lemma 3.** 
*There exists an f:[0,1]→[0,1] such that V(f)=∞ but ${∥f∥}_{W}\leq 2$.*


**Proof.** Let f:[0,1]→[0,1] be the piecewise linear function defined on $x_{n}=\frac{1}{n}$, n≥1, by$$f\left(\frac{1}{n}\right)=\sum_{k=1}^{n}\frac{{\left(-1\right)}^{k+1}}{k}$$
and extended to [0,1] by linear interpolation.Clearly, $V\left(f\right)=\sum_{n=1}^{\infty }\frac{1}{n}=\infty$. To bound ${∥f∥}_{W}$, note that any $x,x^{\prime }\in \left[0,1\right]$ witnessing $\frac{|f\left(x\right)-f\left(x^{\prime }\right)|}{|x-x^{\prime }|}\geq t$ also verify $\left|x-x^{\prime }\right|\leq \frac{1}{t}$. Let $I_{n}$ denote the interval $\left[\frac{1}{n+1},\frac{1}{n}\right]$. If $\Lambda_{f}\left(x\right)\geq t$, then there is an $x^{\prime }$ such that $\frac{|f\left(x\right)-f\left(x^{\prime }\right)|}{|x-x^{\prime }|}\geq t$. Now, either *x* or $x^{\prime }$ lies in $I_{n}$ for n≥t. If $x\in I_{n}$ with n≥t, then $x\leq \frac{1}{t}$. If, however, $x^{\prime }\in I_{n}$ for n≥t, then $x^{\prime }\leq \frac{1}{t}$ and since $\left|x-x^{\prime }\right|\leq \frac{1}{t}$, we have $x\leq \frac{2}{t}$. We conclude that $\Lambda_{f}\left(x\right)\geq t$ implies $x\leq \frac{2}{t}$ and hence $P\left(\Lambda_{f}\left(x\right)\geq t\right)\leq \frac{2}{t}$; this proves the claim. □

Remark.

Another function with this property is $x\sin\frac{1}{x}$.

### 2.3. Proof of Theorem 3

#### 2.3.1. Proof That ${\mathrm{fat}}_{\gamma }\left(F_{W}\right)=\infty$ Whenever $\gamma \leq \frac{L}{6}$

Consider the partition of [0,1] into segments $I_{n}=\left[x_{n+1},x_{n}\right]$ where $x_{n}=2^{-n}$. We define $f\left(x_{n}\right)={\left(-1\right)}^{n}\gamma$. This specifies *f* at all endpoints of $I_{n}$. For $x\in (x_{n+1},x_{n})$, we define $f\left(x\right)={\left(-1\right)}^{n}\frac{\gamma }{|I_{n}|}\left(x-x_{n+1}\right)+{\left(-1\right)}^{n}\frac{\gamma }{|I_{n}|}\left(x-x_{n}\right)$, i.e., *f* is piecewise linear with slope ${\left(-1\right)}^{n}4\gamma 2^{n}$ in $I_{n}$. Similarly to Lemma 3, if $\frac{|f\left(x\right)-f\left(x^{\prime }\right)|}{|x-x^{\prime }|}\geq t$, then $\left|x-x^{\prime }\right|\leq \frac{2\gamma }{t}$. Now, suppose $\Lambda_{f}\left(x\right)\geq t$, i.e., there exists $x^{\prime }$ with $\frac{|f\left(x\right)-f\left(x^{\prime }\right)|}{|x-x^{\prime }|}\geq t$. This implies that either *x* or $x^{\prime }$ lies in $I_{n}$ for some $n\geq \log\frac{t}{4\gamma }$ (the slope of the line connecting *x* and $x^{\prime }$ lies between the slopes of the segments containing $x,x^{\prime }$). If $x\in I_{n}$ for some $n\geq \log\frac{t}{4\gamma }$, then $x\leq x_{n}\leq \frac{4\gamma }{t}$. If, however, $x^{\prime }\in I_{n}$ for some $n\geq \log\frac{t}{4\gamma }$ then $x^{\prime }\leq \frac{4\gamma }{t}$ and since $\left|x-x^{\prime }\right|\leq \frac{2\gamma }{t}$, we have $x\leq \frac{6\gamma }{t}$. Since $\Lambda_{f}\left(x\right)\geq t$ implies $x\leq \frac{6\gamma }{t}$, we can conclude that ${∥f∥}_{W}\leq 6\gamma$. An immediate corollary is that ${\bar{\mathrm{Lip}}}_{L}^{w}$γ-shatters the infinite set ${\left\{x_{n}\right\}}_{n=1}^{\infty }$ for $\gamma \leq \frac{L}{6}$ — which is even stronger than having arbitrarily large γ-shattered sets. Note that this is close to tight, since for $\gamma >\frac{L}{2}$, we cannot γ-shatter even a set of two points. Suppose $f\left(x_{1}\right)>\frac{L}{2}$ and $f\left(x_{2}\right)<-\frac{L}{2}$, then for $x\in [x_{1},x_{2}]$ we have $\Lambda_{f}\left(x\right)>\frac{L}{|x_{2}-x_{1}|}$, hence ${∥f∥}_{W}>L$, which means $\left\{x_{1},x_{2}\right\}$ is not γ-shattered by ${\bar{\mathrm{Lip}}}_{L}^{w}$. □

#### 2.3.2. Proof That
${\mathrm{fat}}_{\gamma }\left(F_{S}\right)=1+\left\lfloor \frac{L}{2\gamma }\right\rfloor$
for γ>0

The upper bound follows immediately from $V\left(f\right)\leq {∥f∥}_{\mathrm{Lip}}$. For the lower bound, take a 2γ/L packing $\left\{x_{1},\ldots ,x_{1+\left\lfloor \frac{L}{2\gamma }\right\rfloor }\right\}$ of [0,1]. For labeling $y_{i}\in {\left\{-1,1\right\}}^{n}$, consider the linear interpolation of $\left(x_{i},y_{i}\gamma \right)$, and observe that the interpolation *f* satisfies $\Lambda_{f}\left(x\right)\leq \frac{2\gamma }{2\gamma /L}=L$ everywhere. □

## Data Availability

No new data were created or analyzed in this study. Data sharing is not applicable to this article.

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
