# Peer review of "Bounded Variation Separates Weak and Strong Average Lipschitz"

_entropy, 2025, doi:10.3390/e27090974_

Round 1

Reviewer 1 Report

Comments and Suggestions for Authors

I have very few comments on this timely and mathematically interesting note.  It is always useful to establish inequalities between metrics, and this papier does exactly that, closing a gap between several Lipschitz-related norms.

Just minor remarks:  the capitalization in the references is off. Also, some sentences are jointed by a comma, which is not allocated in English.

Author Response

Thank you for your suggestions, we have implemented them.

Reviewer 2 Report

Comments and Suggestions for Authors

Report on the paper: "Bounded variation separates weak and strong average Lipschitz"

In this paper the authors give as their main theorem a proof for the containments of strong and weak average Lipschitz classes and compare them to BV function class.

They disscus about this new notions of smoothness of a function (introduced in a previous paper in which one of the authors was involved). To be specific, strong and weak average smoothness based on the local slope of a function. This notions are very relavant since there is a gap between the "smoothness" of a function of bounded variation and a Lipschitz function, and so, this result gives us a clear picture of the containments of this function classes, which is a very interesting result.

The proofs in this paper are correct, as far as I checked. The article is well-written and well-motivated. The results are relavant to the corpus of knowledge and I am certain that these notions of smoothness will be applied and used in many other areas of mathematics, soon.

For the previous mentioned reasons, I am glad to recommend the paper for publication after a very minor correction and also a very minor suggestion.

First there is a typo in line 31: Liptschiz -> Lipschitz.

And, even if it is clear from the context the notation $V_{f}([0,x])$, I recommend to introduced it before it is used in line 81, just for the reader's convenience.

Author Response

(The authors gave the same response as above.)

Reviewer 3 Report

Comments and Suggestions for Authors

This interesting and well-written paper concerns two recently introduced notions regarding real-valued functions defined on metric spaces. Specializing to the real line, the authors compare these notions to the classical notion of bounded variation (BV, for short). The authors show that the weaker notion is strictly weaker than BV while the stronger one is strictly stronger. Since the notion of bounded variation is of fundamental importance in the study of real functions, my recommendation is that (a revised version of) this paper be accepted for publication. When the authors prepare the revised version of their paper, they should take into account the following (rather minor) comments, questions and suggestions. 

(1) Line 5: "notion strictly" ---> "notion is strictly"

(2) Line 23: "as" ---> by"

(3) Line 51: "to to" ---> "to" 

(4) Line 55: What happens when $\gamma > 1/6$? 

(5) Line 67: Please place a comma at the very end of this line. 

(6) Line 78: Please see item (4) above. 

(7) Line 83: "Thus" ---> "Thus the inequality"

(8) Line 83: "implies" ---> "implies that" 

(9) Line 87: "$I$" ---> "I,$"

(10) Line 88: "implies" ---> "implies that"

(11) Line 111: ", then" ---> ". Then"

(12) Line 113: "definition" ---> "the definition"

(13) Line 134: "$n \ge t$" ---> "$n \ge t,$"

(14) Line 142: "on" ---> "at"

(15) Line 155: "interpolation" ---> "interpolating"

(16) Line 157: What could be future research directions in this area?   

Author Response

Thank you for your suggestions, we have implemented them. Regarding (16) -- future directions -- A natural follow-up direction is to extend the ideas of weak/strong average Lipschitzness to broader function spaces. [3,4] define average Hölder smoothness with a modified version of $\Lambda_f(x)$ (it is raised to a constant power, and the weak and strong averages are defined analogously) and shows covering bounds for functions of bounded average Hölder smoothness. We can now ask whether the modified version also coincides with known smoothness measures. Extending our discussion to general metric spaces raises the question of how functions of bounded weak and strong averages relate to Sobolev spaces. HajÅ‚asz, "Sobolev spaces on an arbitrary metric space" (1996) defines Sobolev conditions on general metric spaces. The relationship between our average-based notions of smoothness and these generalizations of classical definitions to arbitrary metric spaces remains unclear.